# Quantifying benefits of the Danish transfat ban for coronary heart disease mortality 1991–2007: Socioeconomic analysis using the IMPACTsec model

**Kirsten Schroll Bjoernsbo**[1*], **Albert Marni Joensen**[2], **Torben Joergensen**[1,3,4],
**Soeren Lundbye-Christensen**[5‡], **Anette Bysted**[6], **Tue Christensen**[6], **Sisse Fagt**[6],
**Simon Capewell**[7], **Martin O'Flaherty**[7]

1 Center for Clinical Research and Prevention, Bispebjerg and Frederiksberg Hospital, Frederiksberg, Denmark, 2 Department of Cardiology, North Denmark Region Hospital, Hjoerring, Denmark, 3 Department of Public Health, Faculty of Health and Medical Sciences, University of Copenhagen, Copenhagen, Denmark, 4 Faculty of Medicine, Aalborg University, Aalborg, Denmark, 5 Unit of Clinical Biostatistics, Aalborg University Hospital, Aalborg, Denmark, 6 National Food Institute, Technical University of Denmark, Kgs. Lyngby, Denmark, 7 Department of Public Health and Policy, University of Liverpool, Liverpool, United Kingdom

☯ These authors contributed equally to this work.
‡ SLC also contributed equally to this work.
* Kirsten.bjoernsbo@regionh.dk

**Data Availability Statement:** All relevant data are within the manuscript and its Supporting information files.

## Abstract

Denmark has experienced a remarkable reduction in CVD mortality over recent decades. The scale of the health contribution from the Danish regulation on industrially produced *trans* fatty acid (ITFA) has therefore long been of interest. Thus the objective was to determine health and equity benefits of the Danish regulation on ITFA content in Danish food, by quantifying the relative contributions of changes in ITFA intake, other risk factors and treatments on coronary heart disease (CHD) mortality decline from 1991 to 2007 in Denmark, stratified by socioeconomic group. To evaluate the effects of the ITFA ban (Danish Order no. 160 of March 2003) the Danish IMPACT$_{SEC}$ model was extended to quantify reductions in CHD deaths attributable to changes in ITFA (%E) intake between 1991–2007. Population counts were obtained from the Danish Central Office of Civil Registration, financial income from Statistics Denmark and ITFA intake from *Dan-MONICA III (1991)* and *DANSDA (2005–2008)*. Participants were adults aged 25–84 years living in Denmark in 1991 and 2007, stratified by socioeconomic quintiles. The main outcome measure was CHD deaths prevented or postponed (DPP). Mean energy intake from ITFA was decimated between 1991 and 2007, falling from 1.1%E to 0.1%E in men and from 1·0%E to 0·1%E in women. Approximately 1,191 (95% CI 989–1,409) fewer CHD deaths were attributable to the ITFA reduction, representing some 11% of the overall 11,100 mortality fall observed in the period. The greatest attributable mortality falls were seen in the most deprived quintiles. Adding ITFA data to the original IMPACTsec model improved the overall model fit from 64% to 73%. In conclusion: Denmark's mandatory elimination of ITFA accounted for approximately 11% of the substantial reduction in CHD deaths observed between 1991 and 2007. The most deprived groups

**Funding:** TJ Creation of the Transfat database was funded by the Danish Health Foundation "Helsefonden 18-B-0355" https://helsefonden.dk/ Helsefonden had no role in study design, data collection and analysis, decision to publish, or preparation of the manuscript.

**Competing interests:** I have read the journal's policy and the authors of this manuscript have the following competing interests: AMJ has given lectures at educational events arranged by Bristol Meier Squibb Denmark, Novartis Healthcare A/S, Bayer A/S This does not alter our adherence to PLOS ONE policies on sharing data and materials.

benefited the most, thus reducing inequalities. Adopting the Danish ITFA regulatory approach elsewhere could substantially reduce CHD mortality while improving health equity.

## Introduction

Cardiovascular diseases are the leading cause of death globally, generating some 18.6 million deaths in 2019 (31% of total deaths) [1]. Consumption of industrially produced *trans* fatty acids (ITFA) is causally associated with increased risk of coronary heart disease (CHD) morbidity and mortality [2]. In 2019, a diet high in ITFA (at least 1% of total energy intake) was estimated to account for approximately 645,000 deaths globally [1]. The detrimental health effects of ITFA have been increasingly recognised since the early 1990s. ITFA substantially increase CHD risk by raising LDL-cholesterol, reducing HDL cholesterol, promoting systemic inflammation and impairing endothelial cell function [2]. On a per-calorie basis, ITFA increase CHD risk more than any other type of fat: every 1% increase in daily energy obtained from ITFA raises CHD mortality by approximately 12% [2]. Removal of ITFA from the global food supply is therefore a key element in WHO's diet strategy [3]. Different policies have been successfully utilized to reduce ITFA across the globe. However, voluntary approaches risk the persistence of high ITFA intake in residual pockets of the population because more deprived communities tending to purchase processed products providing 'cheaper' calories [4]. Conversely, national legislative bans, such as seen originally in Denmark, then Iceland, Austria and Switzerland have proven to be extremely effective, essentially eliminating ITFA in foods [5]. However, further evidence is needed to quantify the health benefits of this legislation, particularly whether it might particularly benefit more deprived socio-economic subgroups with their typically higher ITFA intakes. In Denmark, cardiac mortality has declined dramatically, falling by 74% between 1991 and 2007, equivalent to approximately 11,000 fewer CHD deaths [6]. Joensen et al. 2018 used the IMPACTsec model in a detailed examination of that CHD mortality fall. Improved treatments apparently accounted for about 25% of the mortality decrease, whereas risk factor improvements accounted for substantially more, approximately 40%. However, the IMPACTsec model failed to explain approximately one third of the mortality decline, this gap perhaps reflecting both imprecise data and unquantified risk factors, notably the legislated reduction in ITFA. The size of the contribution from the Danish ITFA ban in 2004 has been much discussed but rarely analysed. Restrepo & Rieger 2016 employed synthetic control methods to estimate that some 700 deaths were prevented annually [7]. However, Restrepo & Rieger lacked detailed ITFA trend data, and did not consider socio-economic aspects. We therefore created a Danish Database to analyse ITFA intake trends, then quantified their potential impact on recent CHD mortality declines in Denmark in different socioeconomic groups, while also considering changes in other risk factors and treatments. Thus, the objective of the present study was to determine health and equity benefits of the Danish regulation on industrially produced *trans* fatty acid (ITFA) content in Danish food, by quantifying the relative contributions of changes in ITFA intake, other risk factors and treatments on coronary heart disease (CHD) mortality decline from 1991 to 2007 in Denmark, stratified by socioeconomic group.

## Materials and methods

### The Danish IMPACTsec ITFA model

The IMPACTsec model is a deterministic, cell-based model, initially developed in UK to help explain the contribution of the change in different risk factors and treatment uptakes on CHD

mortality in different socio-economic groups [8–11]. To evaluate the effects of phasing out ITFA on mortality, life-years and the underlying CHD burden, we extended the Danish IMPACTsec model with energy adjusted ITFA data from two Danish cohorts: *Dan-MONICA III* (Multinational MONItoring of trends and determinants in CArdiovascular disease; in 1991) [12]. and DANSDA (the Danish National Surveys of Dietary Habits and Physical Activity; 2005–2008) [13]. The Danish IMPACTsec ITFA model calculates how much changes in treatments and risk factor prevalence have contributed to the change in CHD mortality in the study period. The model uses information from many different data sources on population and CHD patient numbers, uptake of evidence based medical treatments and case fatality reduction for every medical and surgical treatment currently in use in nine patient groups [10–11]. The model also includes prevalence of risk factors from two time points, 1991 and 2007, and uses mortality risk for all the major CHD risk factors: smoking, systolic blood pressure, total cholesterol, body mass index (BMI), diabetes, physical inactivity.

## Data sources

The original Danish IMPACTsec model examined trends in the Danish population between 1991 and 2007 [6]. Population data were obtained from the Danish Central Office of Civil Registration. That contains complete information on vital status, date of birth and a unique personal 10-digit code which makes it possible to merge information from different registers on an individual level [14]. The population was divided into 60 different strata based on gender, six age-groups (25±34, 35±44, 45±54, 55±64, 65±74 and 75±84 years), and five socioeconomic-level quintiles (based on gender and age-specific personal tax-income information obtained from Statistics Denmark) [6].

ITFA data were calculated from dietary intake registered by 663 men and women (aged 30 to 70 in 1991) from the *Dan-MONICA III* (7day diary) and 2,792 men and women (aged 25–75 in 2005–2008) participating in the DANSDA (7day diary) using the Gies system [15] applying recipes from the ITFA database. Period specific recipes and information on ITFA content in margarines, shortenings, and different foods on the Danish market 1991–2007 were applied. In years without analytical results, the ITFA content in foods was estimated based on the results from the years closest to the year in question, e.g. from the assumption of a linear decrease in the ITFA content during the intervening years. This means that the content for each year was reduced proportionally to the years with known levels. In other cases, the ITFA content in foods was assessed constant for a given period. ITFA intakes were given both as g/day and as %E (percentage of total daily energy intake). Grams of ITFA were transformed to percent energy (%E) using the conversion value of 37 kJ/g [16]. The reduction in CHD deaths for a given reduction in ITFA was based on the meta-analysis by Mozaffarian et al. (2006) [2], using O'Flaherty et al.'s (2012) [17] stratification into gender and age-specific mortality reductions.

*Dan-MONICA III* was approved by the ethics committee of Copenhagen County (KA 90238). The DANSDA surveys are conducted in accordance with the guidelines laid down in the Declaration of Helsinki and are approved by the Danish Data Protection Agency. The Danish National Committee on Health Research Ethics has decided that, according to Danish Law, DANSDA does not require approval. Verbal informed consent was obtained from all participants.

## Deaths prevented or postponed (DPPs)

Mortality data were obtained from The Danish Cause of Death Registry—CHD deaths (1991: International Classification of Diseases (ICD)-8 code 410±414 and 427; 2007: ICD-10 code

I20-25 and I50). The numbers of CHD deaths expected in 2007, had mortality rates remained unchanged since 1991, were calculated by multiplying the age- and sex-specific CHD mortality in 1991 by the relevant population counts for 2007. The difference between the number of observed and expected deaths then represented the decline in CHD mortality, i.e. the total number of CHD deaths prevented or postponed (**DPPs**) requiring to be explained by the combined changes in risk factor levels and treatment uptakes between 1991 and 2007.

## Sensitivity analyses

We used probabilistic sensitivity analysis to estimate the effect of uncertainty in key parameters. We performed 1000 runs of the full model in *R version 3.0.1.41*. The key parameters included consumption of ITFA, link to mortality from coronary heart disease, mortality from coronary heart disease, numbers of patients with coronary heart disease (incidence), the contribution from cholesterol and the strength of the gradient of socioeconomic class (S2a and S2b Table in S1 Appendix). Further details of the model, model assumptions and sensitivity analyses are provided in the S1 Appendix and in the paper by Joensen et al. 2018 [6].

## Results

### ITFA consumption in 1991 (Monica 3) and in 2007 (DANSDA)

MEN. In 1991, mean ITFA in men was 2.9 ± 2.6 g/d (95% CI 0.3–7.6), equivalent to a mean ITFA %E of 1.1 ± 0.8%E (95% CI 0.1–2.6). Mean total energy intake was 10.0 ±2.8 MJ/d (95% CI 5.5–14.8). In 2007, mean ITFA fell to 0.3 ± 0.4 g/d (95% CI 0.0–1.0), representing a mean ITFA of 0.1 ± 0.1%E (95% CI 0.0–0.3) in a mean total energy intake of 10.4 ±2.9 MJ/d (95% CI 6.0–15.5).

WOMEN. In 1991, mean ITFA in women was 2.2 ± 1.6 g/d (95% CI 0.3–5.3), equivalent to a mean ITFA %E of 1.0 ± 0.7%E (95% CI 0.2–2.4). Mean total energy intake of 7.7 ±2.1 MJ/d (95% CI 4.5–11.3). In 2007, mean ITFA in women fell to 0.2 ± 0.4g/d (95% CI 0.0;0.9), representing a mean ITFA of 0.1 ± 0.1%E (95% CI 0.0;0.4) in a total mean energy intake of 8.1 ±2.2 MJ/d (95% CI 4.8;11.8).

ITFA intake (%E) for men and women in the *Dan-MONICA III* (1991) and DANSDA (2005–2008) studies, stratified by socio-economic quintile using income, are illustrated in Fig 1, and detailed in S1 Table in S1 Appendix. A socio-economic gradient for ITFA intake,

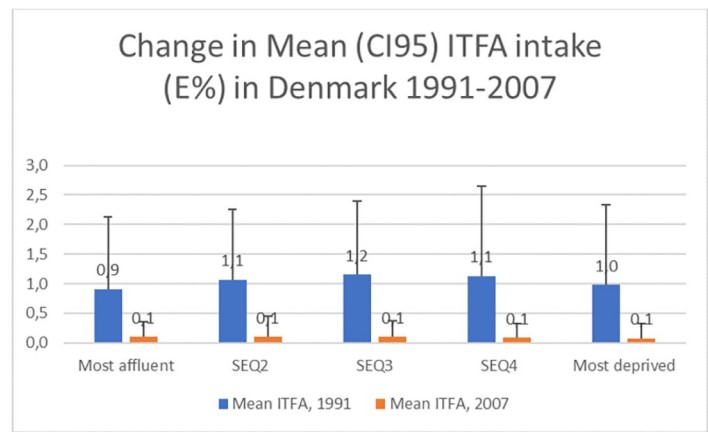

**Fig 1. Trends in mean ITFA intake (%E) in Denmark 1991–2007 divided into socio-economic quintiles.** Top lines indicate 95% CIs.

**Table 1. CHD mortality in 1991 and in 2007, and deaths prevented or postponed in Danish men and women in 2007.**

|  | Men | Women | Total |
|---|---|---|---|
| **Population 1991 (000s)** | 1,555 | 1,647 | 3,202 |
| **Population 2007 (000s)** | 1,700 | 1,760 | 3,461 |
| **Observed CHD deaths 1991** | 8,272 | 5,597 | 13,869 |
| **Observed CHD deaths 2007** | 2,421 | 1,462 | 3,883 |
| **Expected CHD deaths 2007, had 1991 CHD mortality rates remained unchanged** | 9,396 | 5,597 | 14,993 |
| **Deaths Prevented or Postponed (DPPs).** = deaths expected in 2007 minus deaths observed in 2007 | 6,975 | 4,135 | 11,110 |

increasing from the most affluent to the most deprived was apparent in Danes in 1991 (P = 0.0085), but no longer in 2007 (P = 0.3983). The contribution of ruminant TFA intake to energy intake changed little during the period (0.6%E in 1991 and 0.5%E in 2007).

## CHD mortality trends 1991–2007

CHD deaths in 1991 and 2007, and CHD Deaths Prevented or Postponed (DPPs) in Danish men and women in 2007 are summarised in Table 1.

The gap between the CHD deaths observed in 2007 and the number of CHD deaths expected in 2007, had 1991 CHD rates remained, was 11,110 DPPS. The reduction in Danish transfat intake between 1991 and 2007 was estimated to account for approximately 1,190 (95% CI 989–1,409) of those fewer CHD deaths in Denmark in 2007. Thus, reducing ITFA intake explained 11% (95% CI 9%-13%) of the total 11,110 decline in CHD deaths. The ITFA related DPPS were 792 (11%) and 399 (10%) in men and women respectively. The greatest attributable mortality falls were seen in the most deprived quintiles, reflecting their larger reductions in transfat intake. That is, 48% mortality reductions were found in the two most deprived quintiles compared to 30% DPPS in the two most affluent quintiles. Adding ITFA data to the original IMPACTsec model improved the overall fit of the model from 64% to 73% (Table 2).

## Discussion

Cardiovascular disease caused over 6 million premature deaths globally in 2019 [1]. Almost all were preventable, as emphasised by WHO in their Sustainable Development Goal 3.4. That proposed achieving at least a 30% reduction in premature mortality due to noncommunicable disease by 2030 [1]. Legislation to remove ITFA from foods is seen globally as potentially one of the most effective public health approaches for reducing transfat intake and thus reduce noncommunicable diseases [3, 18]. Denmark has experienced a remarkable reduction in CVD

**Table 2. Deaths prevented or postponed in Danish men and women in 2007 attributable to the decrease in ITFA, by socio-economic quintiles.**

|  | ITFA DPPS (95% CI) | % of all DPP generated by each Quintile |
|---|---|---|
| **Most affluent** | 143 (117–171) | 12% |
| **q2** | 220 (182–260) | 18% |
| **q3** | 250 (208–297) | 21% |
| **q4** | 254 (212–302) | 21% |
| **Most deprived** | 324 (271–379) | 27% |
| **Total** | 1,191 (989–1,409) | 100% |
| **Overall Model fit with ITFA** |  | 73% |

mortality over recent decades [6]. The scale of the contribution from the Danish ITFA ban has therefore long been of interest. As part of this strategy, pioneering Danish food policies dramatically cut ITFA consumption by some 90% between 1991 and 2007. This in turn substantially reduced CHD mortality, preventing approximately 1,200 deaths by 2007. That represented some 11% of the total CHD mortality fall observed between 1991 and 2007, (a similar sized contribution of that of decreases in smoking [6]). This analysis does not provide a counterfactual evaluation of the Danish ban itself. That would ideally require a control group or natural experiment approach to better account for general trends and other potential confounding factors. However, the ban likely explains much of the decrease in transfat intake observed over the period, and hence approximately one tenth of the overall fall in cardiovascular disease. The Danish ITFA ban was the culmination of successive interventions to reduce ITFA. The Danish industry had voluntarily replaced transfat in margarines by other fats in the decade before the ban. However, regulation was needed to also eliminate transfats from shortenings and deep-frying fats. ITFA was mainly substituted by saturated fats in bakery goods while frying oils were enriched in monounsaturated fats, resulting in significantly healthier products [5]. As the ITFA intake was already low in 2004, the ITFA regulation was introduced particularly to protect Danes with high intakes of fast food, sweets, and bakery goods. Furthermore, the economic consequences for the industry in Denmark following the ITFA regulation have been limited [19]. Compared to other strategies, the Danish package of voluntary changes by the industry and legislation provides the largest observed reduction in total transfat intake in a population over the period 1976 to 2005 (4.5 g/day) according to Hyseni et.al's systematic review from 2021 [18].

Our study has several strengths. Firstly, the data are based on real ITFA intake measurements—large representative surveys conducted before and after the transfat ban. Secondly, the ITFA contribution is put in perspective, by also quantifying all the other major cardiovascular risk factors and treatments. Thirdly, our analyses are conservative, by only focusing on CHD mortality. ITFA reduction would also decrease other major cardiovascular diseases, such as stroke and heart failure. (Cardiovascular disease mortality in Denmark is 1.5 times higher than CHD mortality alone).

The main limitation of his study is, that TFA harms operate mainly via effects on LDL and HDL cholesterol. If the IMPACTsec Model also separately considers reductions in total cholesterol since 1991, some mortality benefits may have been double counted. However, ITFA reduction decreases CHD risk and mortality via diverse pathways, not just through changes in cholesterol. Furthermore, the potential scale of this potential overestimation can be gauged by our rigorous sensitivity analyses. Thus, even if it were reduced by the entire ITFA 11%, the residual contribution of cholesterol decrease would still explain 23% of the total mortality CHD fall (including both effects of cholesterol lowering drugs (12%) and dietary factors (11%)) (S1 Appendix). Future research would clearly be helpful to further quantify how much of the malign health effects of TFA are explained by cholesterol, and how much by other pathways. Meanwhile, the estimates attributable to decreases in ITFA consumption remain potentially valuable to policy makers in Denmark, and beyond.

Our estimate of about 1,200 CHD DPPs between 1991 and 2007 attributable to ITFA reduction is proportionally consistent with Restrepo and Rieger's estimate of 700 CVD DPPs between the shorter period of 2004 to 2007 [7]. Our longer timescale better captures the substantial reductions in ITFA consumption occurring as the result of cumulative policies pressuring the industry to progressive reformulate and phase out ITFA in most foods in the years preceding the actual legislation [19].

Our IMPACT study is perhaps the first paper to explicitly demonstrate that the greatest benefits were seen in the most deprived groups, reflecting their larger reductions in ITFA

consumption. It suggests that by applying a population-wide approach, equity in transfat intake was achieved. This is comparable with an earlier UK analysis suggesting, that a total UK ban on ITFA in processed foods could prevent or postpone about 7,200 CHD deaths (2.6%) from 2015–20, and particularly benefit the most deprived groups, reducing UK inequalities in CHD mortality by about 15% [20]. Some studies provided lower estimates, likely due to the scenarios modelled and methodologies used. Restrepo & Rieger 2016 employed synthetic control methods to estimate that some 700 deaths were prevented annually [7]. In Argentina, Rubinstein et al. used a comparative risk modelling approach to estimate that a full transfat ban might result in a 1.3%–6.3% reduction in fatal and non-fatal CHD events [21] However, they modelled the effect of transfats on CHD via lipid changes, rather than using the Mozaffarian approach [17].

## Conclusions

CHD mortality in Denmark has decreased by 70% since the 1980s. Over half this mortality fall can be attributed to reductions in major cardiovascular risk factors, but with more affluent socio-economic groups benefiting most. Few expected that labelling, or nutrition education alone would reduce transfat intake for all Danes. But as scientific, social, and political pressures increased, the industry reformulated most margarines and the time for legislation dawned, a structural policy which worked for all—particularly the most deprived. This study is thus a further example of how structural intervention can reduce overall CHD mortality and reduce socio-economic inequalities [22]. Attention is now turning to the other harmful commodities hidden in ultra-processed foods—saturated fats, sugar, and salt. All might benefit from similar regulations. Regulation of industrial transfats in Denmark thus made a substantial contribution to decreasing CHD mortality, with some 11% fewer deaths observed in 2007 compared with 1991. That nation-wide policy ensured that all citizens benefited, especially those in the most deprived groups, thereby reducing inequalities. Furthermore, mandatory regulation represents an important tool for preventing noncommunicable diseases and minimising health inequalities.

The EU implemented the ITFA regulation last year [23]. The Danish ITFA regulatory approach therefore now potentially offers a blueprint for a global strategy to reduce CHD mortality. Similar regulatory strategies should therefore now be considered worldwide to protect consumers from other harmful nutrients such as saturated fats, sugar, and salt.

## Supporting information

**S1 Appendix. Technical appendix for the Danish IMPACT_{SEC} ITFA model including tables on ITFA consumption and sensitivity analyses.**
(DOCX)

## Acknowledgments

We thank Ulla Toft for originally suggesting this Danish transfat analysis. We also thank our UK colleagues for their work in helping develop the original IMPACTsec model, notably Rosalind Raine, Sean Sholes, Madhavi Bajekal (who ensured the integrity of inputs and outputs and provided socio-economic-related methodological solutions), Hande Love (who set up the worksheet template) and Nat Hawkins (who provided support, clinical and therapeutic expertise).

## Author Contributions

**Conceptualization:** Kirsten Schroll Bjoernsbo, Albert Marni Joensen, Torben Joergensen, Simon Capewell, Martin O'Flaherty.

**Data curation:** Kirsten Schroll Bjoernsbo, Albert Marni Joensen, Soeren Lundbye-Christensen, Anette Bysted, Tue Christensen, Sisse Fagt, Martin O'Flaherty.

**Formal analysis:** Kirsten Schroll Bjoernsbo, Albert Marni Joensen, Soeren Lundbye-Christensen, Martin O'Flaherty.

**Methodology:** Simon Capewell, Martin O'Flaherty.

**Writing – original draft:** Kirsten Schroll Bjoernsbo.

**Writing – review & editing:** Albert Marni Joensen, Torben Joergensen, Anette Bysted, Tue Christensen, Sisse Fagt, Simon Capewell, Martin O'Flaherty.

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
