## [Decision Letter · Decision Letter 0]

26 May 2022

PONE-D-22-04093Quantifying benefits of the Danish Transfat Ban for coronary heart disease mortality 1991-2007: socioeconomic analysis using the IMPACTsec modelPLOS ONE

Dear Dr. Bjoernsbo,

Thank you for submitting your manuscript to PLOS ONE. After careful consideration, we feel that it has merit but does not fully meet PLOS ONE’s publication criteria as it currently stands. Therefore, we invite you to submit a revised version of the manuscript that addresses the points raised during the review process.

We look forward to receiving your revised manuscript.

Kind regards,

Paolo Piras

Academic Editor

PLOS ONE

Journal Requirements:

Additional Editor Comments:

"I have read the journal's policy and the authors of this manuscript have the following competing interests:

AMJ has given lectures at educational events arranged by Bristol Meier Squibb Denmark, Novartis Healthcare A/S, Bayer A/S"

Dear authors

I apologize for the delay but I found increible difficulties in finding reviewers.

I attach here the comment of the sole reviwer I found. Please address his/her comments and submit a revised version. On my side..I can say that I found a lack of clarity in the exposition of statistical methods adopted in the paper. For example the MPACTsec model is not explained in full and the reader is referred to some references.

Reviewers' comments:

Reviewer's Responses to Questions

**Comments to the Author**

1. Is the manuscript technically sound, and do the data support the conclusions?

Reviewer #1: Yes

2. Has the statistical analysis been performed appropriately and rigorously? 

Reviewer #1: I Don't Know

3. Have the authors made all data underlying the findings in their manuscript fully available?

Reviewer #1: Yes

4. Is the manuscript presented in an intelligible fashion and written in standard English?

Reviewer #1: Yes

5. Review Comments to the Author

Reviewer #1: I enjoyed reading the paper. Objectives, execution and interpretation are very good. Data sources are pertinent. I only have one minor comment (at the discretion of authors and editors):

In my view the paper does not perform a counterfactual evaluation of the Danish ban itself. While the ban has likely reduced trans fat intake/exposure, there are general trends and other confounding factors that may have contributed to trans fat declines and associated health improvements over the period. This would require a control group or natural experiment approach.

6. PLOS authors have the option to publish the peer review history of their article (what does this mean?). If published, this will include your full peer review and any attached files.

Reviewer #1: No

---

## [Author Response · Author response to Decision Letter 0]

1 Jul 2022

Reviewer #1: I enjoyed reading the paper. Objectives, execution and interpretation are very good. Data sources are pertinent. I only have one minor comment (at the discretion of authors and editors):

In my view the paper does not perform a counterfactual evaluation of the Danish ban itself. While the ban has likely reduced trans fat intake/exposure, there are general trends and other confounding factors that may have contributed to trans fat declines and associated health improvements over the period. This would require a control group or natural experiment approach.

Thank you. We agree that one would ideally want a control group. 

However, that is challenging when considering comparisons between countries. 

Our amended manuscript now 

 a) highlights the substantially larger transfat reduction achieved in Denmark compared with other countries in the Limitations paragraph of the manuscript Discussion.

“Compared to other strategies, the Danish package of voluntary changes by the industry and legislation provides the largest observed reduction in total transfat intake in a population over the period 1976 to 2005 (4.5 g/day) according to Hyseni et.al’s systematic review from 2021 [18]”.

 b) Acknowledges this issue in the Limitations paragraph of the manuscript Discussion.

"This analysis does not provide a counterfactual evaluation of the Danish ban itself. That would ideally require a control group or natural experiment approach to better account for general trends and other potential confounding factors. However, the ban likely explains much of the decrease in transfat intake observed over the period, and hence approximately one tenth of the overall fall in cardiovascular disease."

---

## [Editor Report · Decision Letter 1]

26 Jul 2022

Quantifying benefits of the Danish Transfat Ban for coronary heart disease mortality 1991-2007: socioeconomic analysis using the IMPACTsec model

PONE-D-22-04093R1

Dear Dr. %Schroll Bjoernsbo%,

We’re pleased to inform you that your manuscript has been judged scientifically suitable for publication and will be formally accepted for publication once it meets all outstanding technical requirements.

Kind regards,

Paolo Piras

Academic Editor

PLOS ONE
---

## [Editor Report · Acceptance letter]

9 Aug 2022

PONE-D-22-04093R1 

Quantifying benefits of the Danish Transfat Ban for coronary heart disease mortality 1991-2007: socioeconomic analysis using the IMPACTsec model 

Dear Dr. Bjoernsbo:

I'm pleased to inform you that your manuscript has been deemed suitable for publication in PLOS ONE. Congratulations! Your manuscript is now with our production department. 

Kind regards, 

on behalf of

Dr. Paolo Piras 

Academic Editor

PLOS ONE